# Spatio-Temporal Data Fusion for Satellite Images Using Hopfield Neural Network

**Che Heng Fung [1], Man Sing Wong [1,2,*] and P. W. Chan [3]**

[1]  Department of Land Surveying and Geo-informatics, The Hong Kong Polytechnic University, Kowloon, Hong Kong, China
[2]  Research Institute for Sustainable Urban Development, The Hong Kong Polytechnic University, Kowloon, Hong Kong, China
[3]  The Hong Kong Observatory, Hong Kong, China
*  Correspondence: lswong@polyu.edu.hk

**Abstract:** Spatio-temporal data fusion refers to the technique of combining high temporal resolution from coarse satellite images and high spatial resolution from fine satellite images. However, data availability remains a major limitation in algorithm development. Existing spatio-temporal data fusion algorithms require at least one known image pair between the fine and coarse resolution image. However, data which come from two different satellite platforms do not necessarily have an overlap in their overpass times, hence restricting the application of spatio-temporal data fusion. In this paper, a new algorithm named Hopfield Neural Network SPatio-tempOral daTa fusion model (HNN-SPOT) is developed by utilizing the optimization concept in the Hopfield neural network (HNN) for spatio-temporal image fusion. The algorithm derives a synthesized fine resolution image from a coarse spatial resolution satellite image (similar to downscaling), with the use of one fine resolution image taken on an arbitrary date and one coarse image taken on a predicted date. The HNN-SPOT particularly addresses the problem when the fine resolution and coarse resolution images are acquired from different satellite overpass times over the same geographic extent. Both simulated datasets and real datasets over Hong Kong and Australia have been used in the evaluation of HNN-SPOT. Results showed that HNN-SPOT was comparable with an existing fusion algorithm, the spatial and temporal adaptive reflectance fusion model (STARFM). HNN-SPOT assumes consistent spatial structure for the target area between the date of data acquisition and the prediction date. Therefore, it is more applicable to geographical areas with little or no land cover change. It is shown that HNN-SPOT can produce accurate fusion results with >90% of correlation coefficient over consistent land covers. For areas that have undergone land cover changes, HNN-SPOT can still produce a prediction about the outlines and the tone of the features, if they are large enough to be recorded in the coarse resolution image at the prediction date. HNN-SPOT provides a relatively new approach in spatio-temporal data fusion, and further improvements can be made by modifying or adding new goals and constraints in its HNN architecture. Owing to its lower demand for data prerequisites, HNN-SPOT is expected to increase the applicability of fine-scale applications in remote sensing, such as environmental modeling and monitoring.

**Keywords:** spatio-temporal data fusion; Hopfield neural network; satellite images

## 1. Introduction

Satellite remote sensing for monitoring the geophysical environment has developed rapidly in recent decades. However, earth observation is often constrained by the spatial and temporal resolutions of the satellite images. For example, Landsat satellites capture images with moderate spatial

resolution (30 m) but with a long revisit time of 16 days. When cloud contamination is considered, only 65% of images are adequately cloud-free to be used for earth observation [1]. Hence, the temporal resolution of usable Landsat images is relatively low. In contrast, the MODerate resolution Imaging Spectroradiometer (MODIS) can provide daily images, but at coarser spatial resolution of 250 m, 500 m, and 1 km. In order to utilize different data acquisition capabilities, numerous studies have been conducted to integrate the spatial and temporal resolutions, known as spatio-temporal fusion [2–12]. With the integration of high spatial resolution and high temporal resolution, information provided by different satellites can be better utilized, especially for applications requiring both frequent coverage and high spatial details, such as diurnal reflectance analysis and 24-hour temperature monitoring [13]. These applications can help in environmental modeling and monitoring, and can further assist in making decisions about urban planning [2].

Among existing spatio-temporal data fusion algorithms, the spatial and temporal adaptive reflectance fusion model (STARFM) is the first model developed [3] and it has been widely applied for monitoring environmental changes [5,14]. STARFM was originally designed to fuse the reflectance data of Landsat (at 30 m resolution) and MODIS (at 500 m resolution). It uses one known pair of Landsat and MODIS images and one MODIS image at the prediction date. STARFM assumes that for a pure coarse pixel where only one land cover type exists, the changes in fine pixels within that coarse pixel can be implied directly by the changes in that coarse pixel. For heterogeneous coarse pixels where there are two or more land cover types, a weighted function is used for prediction, which assigns higher weights to the neighboring fine pixels where they are physically closer and spectrally similar to the coarse pixels. Subsequently, several algorithms have been developed to enhance STARFM. For example, the spatial temporal adaptive algorithm for mapping reflectance change (STARRCH) attempts to capture episodic events but more input images are required by this algorithm [5]. The enhanced STARFM, namely ESTARFM, improves the fusion results in heterogeneous areas by applying the linear spectral unmixing (LSU) theory [15] and the concept of conversion coefficient [11]. The spatio-temporal adaptive data fusion algorithm for temperature mapping (SADFAT) was developed to fuse land surface temperature (LST) between Landsat TM and MODIS to produce a daily LST product at 120 m spatial resolution [8]. A fusion model was then developed combining the extreme learning machine (ELM) algorithm and SADFAT to generate 30 m LST products from Landsat ETM+ TIR and MODIS data [2]. Zhu et al. (2016) developed the flexible spatiotemporal data fusion (FSDAF) algorithm using one known pair of fine and coarse images to fuse reflectance data [12] In FSDAF, the intermediate result of thin plate spline (TPS) interpolation is used to assist residual distribution. Therefore, it should have the ability to capture land cover change and episodic events.

Another research area of spatio-temporal data fusion is dictionary-pair learning [6,7]. The sparse-representation-based spatio-temporal reflectance fusion model (SPSTEM) establishes structural correspondences between fine resolution images and the corresponding coarse images for a given area [6]. In SPSTEM, two dictionaries are first trained using the fine and coarse resolution differencing image patches at two known dates, by adopting an optimization equation based on sparse representation and sparse coding. Then the fine resolution differencing image patches associated with the prediction date are estimated by applying the same sparse representation coefficients of the low-resolution image patches. SPSTEM was later modified to adapt only one fine and coarse image pair for the spatio-temporal data fusion. This was done by introducing a two-layer fusion framework, which first predicted the target image at medium resolution, followed by the desired fine resolution, to tackle the problem of large scale difference between fine and coarse images [7]. In order to tackle the drawback of insufficient prior knowledge for sparse representation such as cluster and joint structural sparsity [16] developed an optimization model for spatio-temporal data fusion using semi-coupled dictionary learning and structural sparsity.

Recently, a spatio-temporal data fusion model using deep convolutional neural networks (CNNs) was developed [17]. Two five-layer CNNs are constructed to firstly learn the mapping relationship between MODIS and the low-spatial-resolution (LSR) Landsat images (the first CNN), and then to learn

the super-resolution from the LSR Landsat images to the Landsat images with the original resolution (the second CNN). A training stage and a prediction stage are incorporated in this fusion model.

Although several methods have been developed about spatio-temporal data fusion, there are still some limitations due to stringent data requirements. The CNN fusion algorithm requires abundant data for training the relationship between fine and coarse satellite images. Some fusion algorithms [2,5,6,8,11] require at least two known fine and coarse image pairs, one before and one after the prediction date. This prerequisite prohibits near real-time applications, e.g., to derive real-time fine resolution satellite images based on real-time coarse images from geostationary satellites. In such applications, only images before the prediction time are known and therefore fusion algorithms requiring at least two known image pairs cannot be used. Although fusion algorithms requiring only one known image pair have recently been developed [3,7,12,16], they rely on the spatial correspondence between the fine and coarse resolution images. This only limits the fusion models to satellite platforms which have the same overpass times. However, in real applications, satellite images may not have the same overpass times and hence this poses a great challenge to the issue. In this paper, a new spatio-temporal data fusion algorithm, the Hopfield Neural Network SPatiotempOral daTa fusion model (HNN-SPOT) is developed, having the ability to use one fine resolution image at an arbitrary date and one coarse resolution image at the prediction date, to predict the fine resolution image at the prediction date using the optimization concept in the Hopfield neural network (HNN). An important merit of the HNN-SPOT is that it enables image fusion for satellites crossing the same geographic area with different overpass times. This method has a different concept to the existing one, as HNN-SPOT derives a synthesized fine resolution image for the coarser-resolution satellite at the prediction date, whereas existing methods derive a synthesized image for the finer-resolution satellite at the prediction date by utilizing the spatial relationship between the coarse and fine spatial resolution images at the known date. The difference of this new concept is depicted in Figures 1 and 2.

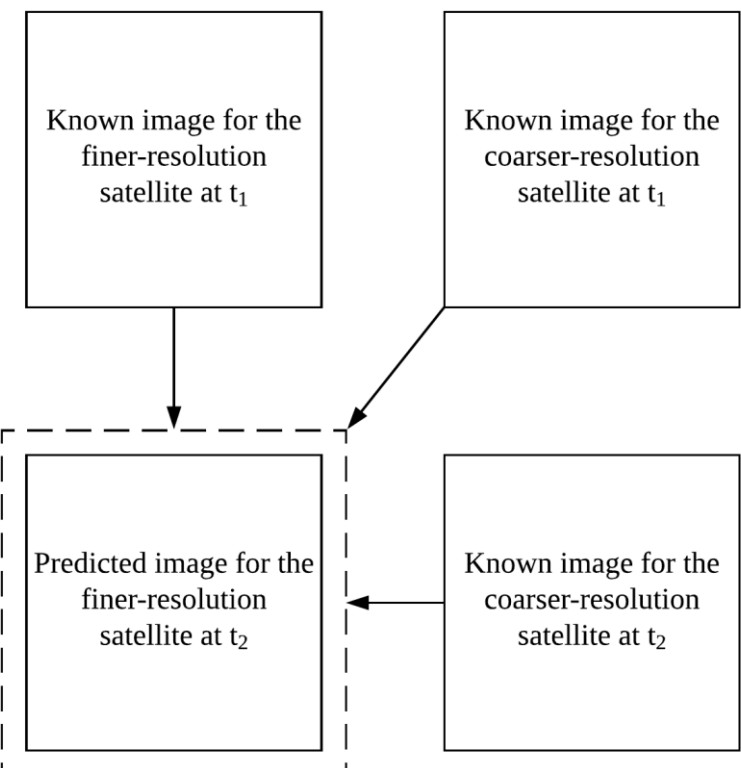

**Figure 1.** Architecture of traditional spatio-temporal data fusion. Suppose $t_1$ represents the known date and $t_2$ represents the prediction date.

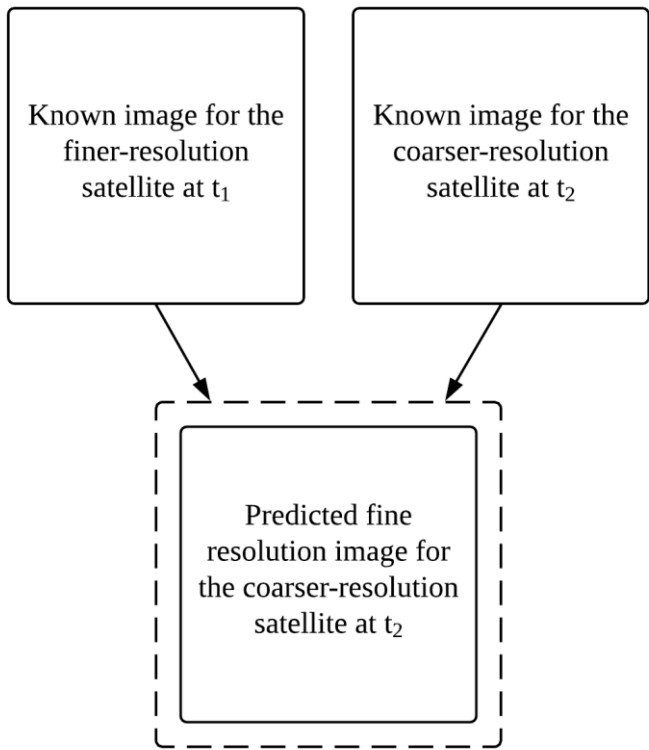

**Figure 2.** Architecture of HNN-SPOT, the newly proposed spatio-temporal data fusion model. Suppose $t_1$ represents the known date and $t_2$ represents the prediction date.

## 2. Methodology

HNN is a type of recurrent artificial neural network and it is shown that it can always converge to a stable or equilibrium state with the use of symmetric weights of each neuron pair without self-connection [18]. This convergence property can therefore be used in energy minimization problems where the energy is minimized at the equilibrium state [19]. In real world problems, an energy can be defined as the summation of the goal to be achieved and the constraints to be followed [20]:

$$Energy = Goal + Constraints. \tag{1}$$

HNN can then be used as a minimization tool to find the minimum energy for which the goal and the constraints are balanced. In remote sensing, HNN has been used in applications of feature identification/tracking; for example, for ice, clouds, and ocean currents [21], and for super-resolution target identification and mapping [20,22]. In this paper, HNN-SPOT is developed, and it further demonstrates in the process of spatio-temporal data fusion, for enhancing the spatial resolution of coarse resolution images.

In HNN-SPOT, one fine resolution image at an arbitrary date (noted as $F_{t1}$) and coarse resolution image at the prediction date (noted as $C_{t2}$) are used to derive the fine resolution image at the prediction date (noted as $F_{t2}$). Intrinsically, the fusion model strikes a balance between the spatial details in $F_{t1}$ and the spectral response value in $C_{t2}$, and therefore the energy function can be literally expressed as:

$$Energy = Spatial\ details\ in\ fine\ resolution + Spectral\ response\ in\ coarse\ resolution. \tag{2}$$

In order to model the above energy function, the network architecture should firstly be well-defined. Thus, each pixel in the fine resolution space is deemed as a neuron and each neuron is identified by its location at the $i$th row and the $j$th column, denoted as $u_{ij}$. Each neuron in the fine resolution space lies in its corresponding coarse pixel at the $x$th row and the $y$th column in the coarse resolution space and

there are $S^2$ neurons in each coarse pixel for the scale factor of $S$ between the fine and coarse resolution images (Figure 3). The network is then initialized with the pixel values in $F_{t1}$ as the initial state of each neuron.

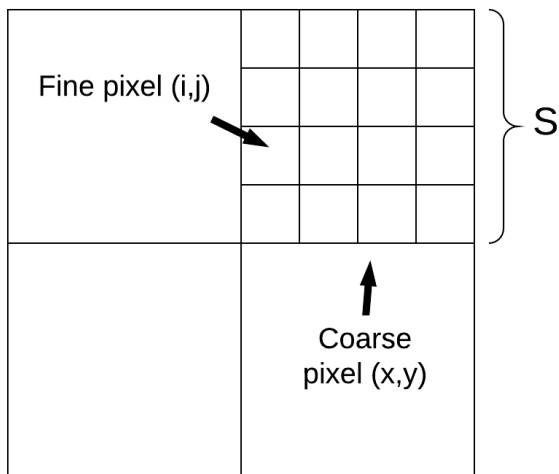

**Figure 3.** Illustration of pixels in HNN-SPOT.

In spatio-temporal image fusion, due to the fact that the minimum energy of the HNN is unknown, it should be implemented with the view of the neuron's motion. From the Euler method, the neuron values can be simulated as in Equation (3) [20]:

$$u_{ij}(t+dt) = u_{ij}(t) + \frac{du_{ij}(t)}{dt}dt, \tag{3}$$

where $dt$ is the time step and $\frac{du_{ij}}{dt}$ is the rate of change of neuron input, which is the negative of the equation of motion in HNN [20]:

$$\frac{du_{ij}}{dt} = -\frac{dE_{ij}}{dv_{ij}}, \tag{4}$$

where $E_{ij}$ is the energy and $v_{ij}$ is the neuron output. $\frac{dE_{ij}}{dv_{ij}}$ can then be estimated as:

$$\frac{dE_{ij}}{dv_{ij}} = k_1\frac{dSpat_{ij}}{dv_{ij}} + k_2\frac{dSpec_{ij}}{dv_{ij}}, \tag{5}$$

where $k_1$ and $k_2$ are the weights for the two energy parameters, $\frac{dSpat_{ij}}{dv_{ij}}$ describes the action for a neuron considering the spatial details in $F_{t1}$ and $\frac{dSpec_{ij}}{dv_{ij}}$ describes the action for a neuron considering the spectral response in $C_{t2}$. In HNN-SPOT, the term $\frac{dSpat_{ij}}{dv_{ij}}$ is formulated through the use of correlation coefficient (see Equation (6)) between the input fine resolution image $F_{t1}$ and the predicted fine resolution image $F_{t2}$. In order to retain the spatial details, the neuron output is arranged such that the correlation coefficient between $F_{t1}$ and $F_{t2}$ is identical, with the assumption of no land cover change:

$$\frac{dSpat_{ij}}{dv_{ij}} = -\frac{1}{2}\left(1 - \tanh\left(r_{ij,w} - thres_r\right)\lambda\right)\left(F1_{ij} - \frac{\sum_{m=i-w}^{i+w}\sum_{n=j-w}^{j+w}F1_{mn}}{(2w+1)^2} + \frac{\sum_{m=i-w}^{i+w}\sum_{n=j-w}^{j+w}v_{mn}}{(2w+1)^2} - v_{ij}\right), \tag{6}$$

where *tanh* is a mathematical function resulting an output value bounded between $-1$ and $1$ (illustrated in Figure 4), $w$ is the half window size for the window where the correlation coefficient is computed, $r_{ij,w}$ is the correlation coefficient between $F_{t1}$ and $F_{t2}$ centered at location $(i, j)$ with half window size $w$, $thres_r$ is the threshold that the correlation coefficient is expected between $F_{t1}$ and $F_{t2}$, $\lambda$ is the gain that

controls the steepness of the *tanh* function (larger $\lambda$ means a steeper *tanh*), and $F1_{ij}$ is the pixel value of $F_{t1}$ at location $(i, j)$. In Equation (6), if the neurons in a certain window fail to satisfy the threshold condition of correlation coefficient (the first part of the equation, i.e., $-\frac{1}{2}\left(1 - \tan h\left(r_{ij,w} - thres_r\right)\lambda\right)$), then the center neuron is modified to enhance the correlation coefficient (the second part of the equation,

i.e., $\left(F1_{ij} - \dfrac{\sum_{m=i-w}^{i+w}\sum_{n=j-w}^{j+w} F1_{mn}}{(2w+1)^2} + \dfrac{\sum\limits_{m\,=\,i\,-\,w}^{i+w}\sum\limits_{n\,=\,j\,-\,w}^{j+w} v_{mn}}{(2w+1)^2} - v_{ij}\right)$).

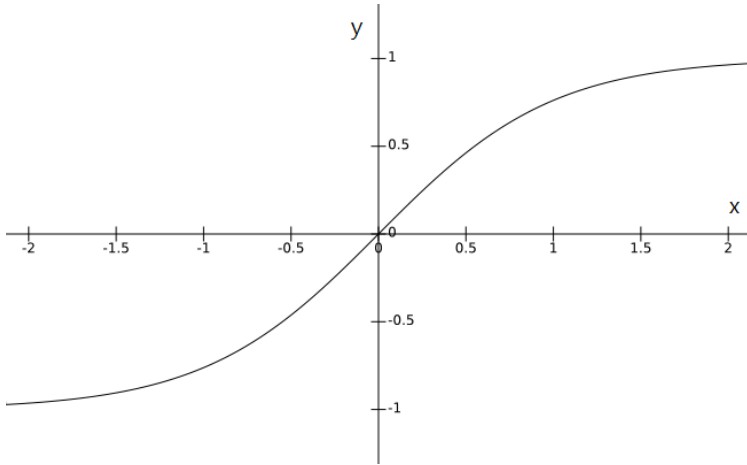

**Figure 4.** The graph of y = tanh(x).

The term $\dfrac{dSpec_{ij}}{dv_{ij}}$ can be modeled using the spectral response in $C_{t2}$. Assuming that a coarse pixel represents the mean value of the fine pixels belonging to it:

$$\frac{dSpec_{ij}}{dv_{ij}} = \frac{1}{S^2}\sum_{m\,\in\,x}\sum_{n\,\in\,y} v_{mn} - C2_{xy}, \qquad (7)$$

where $C2_{xy}$ is the pixel value of $C_{t2}$ at location $(x, y)$. With Equation (7), the neuron output will increase in the situation that a coarse pixel value is greater than the mean of its fine pixels, and vice versa.

To balance the fine spatial details and the coarse spectral responses in the data fusion problem, Equation (3) will implement iteratively until the mean of the absolute difference between the current and the previous neuron output is less than or equal to a small value, $\varepsilon$:

$$\sum_i\sum_j \frac{\left|u_{ij}(t+dt) - u_{ij}(t)\right|}{u_{ij}(t)} / \text{ij} \le \varepsilon. \qquad (8)$$

After converging to a stable state for the HNN algorithm, the preliminary (first round) fusion result would normally be generated. However, this preliminary result contains a blocking effect due to the formation in Equation (7), which is a block-based operation, and therefore the second-round of the HNN algorithm will be operated to tackle this problem. In the second-round operation, the initial neuron state is set to the pixel values of $F_{t1}$ again. However, the first-round result is considered as the coarse resolution image at the prediction date instead of $C_{t2}$. The other part of the calculation process remains the same as in the first-round operation except for Equation (7), which is modified as:

$$\frac{dSpec_{ij}}{dv_{ij}} = \frac{\sum\limits_{m\,=\,i\,-\,w}^{i+w}\sum\limits_{n\,=\,j\,-\,w}^{j+w} v_{mn}}{(2w+1)^2} - \frac{\sum\limits_{m\,=\,i\,-\,w}^{i+w}\sum\limits_{n\,=\,j\,-\,w}^{j+w} F2\_1st_{mn}}{(2w+1)^2} \quad \left(for\ 2^{nd}\ round\right), \qquad (9)$$

where $F2\_1st_{mn}$ is the pixel value of the predicted image of $F_{t2}$ after the first-round operation at location $(m, n)$. Equation (9), compared to Equation (7), distributes the spectral information inside each moving

window rather than each coarse image block. This formulation assumes that the spectral response has been predicted in an acceptable level for $F_{t2}$ in the first-round operation (although there is a blocking effect) and therefore the first-round spectral information can be used to guide the spectral information distribution in the second-round HNN operation without causing the blocking effect.

Through the mentioned HNN-based architecture, the satellite image at fine resolution can be predicted using only two images: one fine image at an arbitrary date and one coarse image at the prediction date. However, it should be noted that, unlike other existing fusion algorithms that require at least one known fine-coarse image pair at the known date, HNN-SPOT predicts the fine resolution image for the satellite images with coarser resolution while other algorithms predict another fine resolution image for the satellite with finer resolution (see Figures 1 and 2).

## 3. Evaluation of Algorithm Performance

To evaluate the HNN-SPOT algorithm, simulated datasets and real datasets were prepared. The simulated dataset was prepared for the purpose of assessing the theoretical accuracy of the fusion algorithm, where only a scale difference exists between the fine and coarse resolution images. Thus, the coarse resolution images at the prediction dates were simulated by upscaling the fine resolution satellite images at the prediction dates using the nearest-neighbor approach. Three simulated datasets were prepared with different satellite images and with different scale factors (from scale factor of 3 to 16). The first site selected is Lantau Island together with the International Airport of Hong Kong. In this study area, the fusion of Landsat–MODIS satellite images was simulated with scale factor of 16. As observed from the Landsat 8 images collected on 31 December 2013 and 18 October 2015, there is a newly built reclamation island near the airport. The second site is located at the Hong Kong Island. The fusion between Sentinel 2 and Landsat images was simulated with scale factor of 3. There is a remarkable change for the vegetation pattern in the racecourse when comparing the Sentinel 2 satellite images collected on 1 January 2016 and 14 February 2017. The last simulated dataset is located at the Werribee South in Victoria, Australia where there was remarkable land cover change due to agricultural practices. Fusion was performed between PlanetScope and Landsat images (scale factor = 10) using two PlanetScope images on 31 May 2018 and 22 June 2018 (Table 1).

**Table 1.** Details of study areas for simulated datasets.

| Study Area | Site Names | Descriptions on Land Cover | Satellite Combinations (Quantity being Fused) | Dates of Data Acquisition |
|---|---|---|---|---|
| 1 | Lantau Island and International Airport (Hong Kong, China) (22.2632°N, 113.9489°E) | Land cover with vegetation, manmade structures and sea | Landsat visible at 30 m + resampled Landsat | 31 December 2013 |
| | | A new artificial island was built near the airport | (MODIS-like) visible at *480 m (reflectance) | 18 October 2015 (P) |
| 2 | Hong Kong Island (Hong Kong, China) (22.2618°N, 114.1815°E) | Land cover with vegetation and urban structures | Sentinel-2 visible at 10 m + resampled | 1 January 2016 |
| | | Remarkable vegetation change was observed for the racecourse | Sentinel-2 (Landsat-like) visible at 30 m (reflectance) | 14 February 2017 (P) |
| 3 | Werribee South (Victoria, Australia) (37.9328°S, 114.6997°E) | Farmland with significant tonal change | PlanetScope visible at 3 m + resampled | 22 June 2018 |
| | | | PlanetScope (Landsat-like) visible at 30 m (radiance) | 31 May 2018 (P) |

**Remarks:** P: Prediction date; *480 m: Spatial resolution of 480 m, instead of 500 m, was used for the MODIS-like image for achieving integral scale factor in data fusion.

In the simulated datasets, only the scale factor differs between the fine resolution and coarse resolution images (Figure 5). However, in real data fusion, there are more problems including inconsistency and misalignment between different satellite sensors, i.e., differences between the viewing angles, solar geometries, and spectral characteristics of the sensors used. In addition, registering different satellite images to the same coordinate system may introduce geometric error. Therefore, the HNN-SPOT algorithm was also tested with the real dataset to demonstrate its applicability to real

world applications (Figure 6). Two study sites in Australia were selected to test the fusion algorithm for their spatial and temporal characteristics respectively [23]. The tests apply to real data fusion between Landsat and MODIS reflectance with scale factor of 16. The site Coleambally irrigation area is considered as spatially heterogeneous due to crop phenology differences over the irrigation area. The Landsat reflectance on 2 November 2001 and MODIS reflectance (resampled to 480 m using nearest-neighbor method) on 27 April 2002 were collected for running the HNN-SPOT algorithm. The second study site—the Lower Gwydir Catchment—is considered as temporally dynamic since there was a flood in mid-December 2004. The Landsat reflectance on 26 November 2004 and MODIS reflectance (resampled to 480 m using nearest neighbor method) on 12 December 2004 were collected by applying the HNN-SPOT algorithm (Table 2).

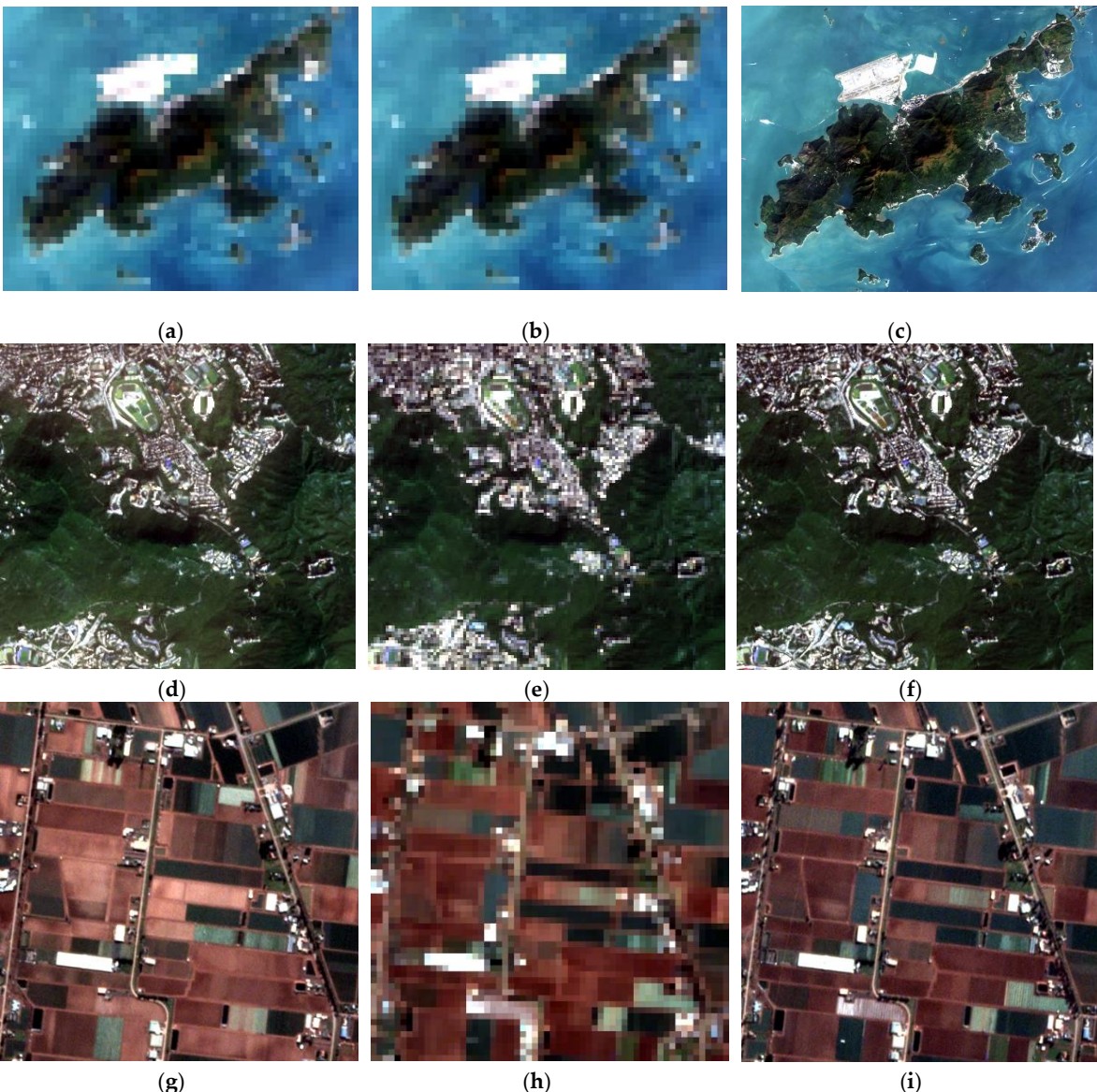

**Figure 5.** Simulated datasets: (**a**): Landsat 8 on 12/31/2013 (30 m) for study area 1; (**b**): Resampled Landsat 8 on 10/18/2015 (480 m) for study area 1; (**c**): Landsat 8 on 10/18/2015 (30 m) for study area 1; (**d**): Sentinel-2 on 1/1/2016 (10 m) for study area 2; (**e**): Resampled Sentinel-2 on 2/14/2017 (30 m) for study area 2; (**f**): Sentinel-2 on 2/14/2017 (10 m) for study area 2; (**g**): PlanetScope on 6/22/2018 (3 m) for study area 3; (**h**): Resampled PlanetScope on 5/31/2018 (30 m) for study area 3; (**i**): PlanetScope on 5/31/2018 (3 m) for study area 3.

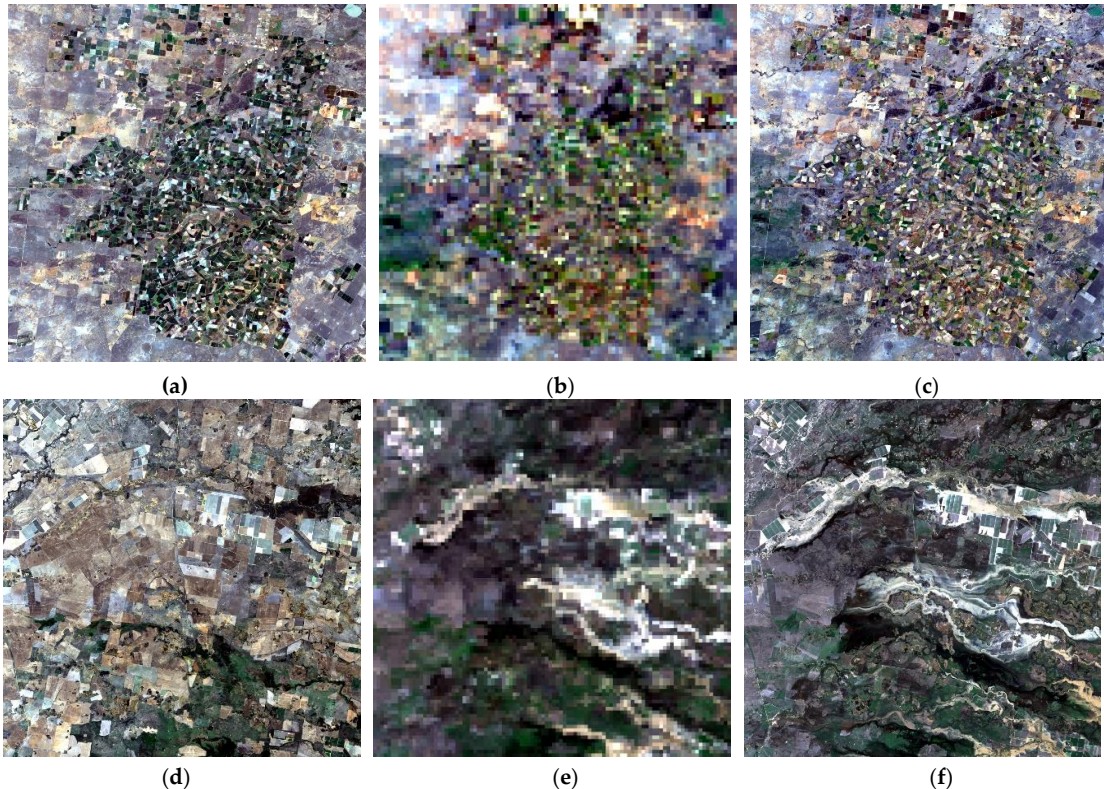

**Figure 6.** Real datasets: (**a**): Landsat on 11/2/2001 (30 m) for study area 4; (**b**): MODIS on 4/27/2002 (480 m) for study area 4; (**c**): Landsat on 4/27/2002 (30 m) for study area 4; (**d**): Landsat on 11/26/2004 (30 m) for study area 5; (**e**): MODIS on 12/12/2004 (480 m) for study area 5; (**f**): Landsat on 12/12/2004 (30 m) for study area 5.

**Table 2.** Details of study areas for real datasets.

| Study Area | Site Names | Descriptions on Land Cover | Satellite Combinations (Quantity being Fused) | Dates of Data Acquisition |
|---|---|---|---|---|
| 4 | Coleambally Irrigation Area (New South Wales, Australia) (34.9069°S, 145.9546°E) | Crop phenology in irrigation area | Landsat visible at 30 m + resampled MODIS visible at 480 m (reflectance) | 2 November 2001 <br><br> 27 April 2002 (P) |
| 5 | Lower Gwydir Catchment (North New South Wale, Australia) (29.1604°S, 149.2694°E) | Flooding in mid-December, 2004 | Landsat visible at 30 m + resampled MODIS visible at 480 m (reflectance) | 26 November 2004 <br><br> 12 December 2004 (P) |

**Remarks:** P: Prediction date.

To achieve compatibility in setting the parameters in the HNN-SPOT, for all cases being studied, the window size was set to be the same as the scale factor between the fine and coarse images. The weights $k_1$ and $k_2$ were both set as 1 for assuring equal importance of achieving spatial detail in $F_{t1}$ and spectral response in $C_{t2}$. *thres$_r$* was defined as 1 with the assumption that the detailed spatial structure in $F_{t1}$ remains unchanged when achieving the goal of spatial detail. $\lambda$ was defined as 100, a practically usual value [22]. $\varepsilon$ was set to a universal value of 1% indicating the convergence of the HNN, and the use of the unit of percentage can assure the fusion ability in different types of quantities. The influence of different parameter settings to the performance of the HNN-SPOT algorithm will be discussed in detail in Section 5. The STARFM algorithm was also programmed as a benchmark in evaluating the HNN-SPOT algorithm for its popularity and acceptability in the field of spatio-temporal data fusion. Some other advanced data fusion algorithms were not evaluated in this paper since the HNN-SPOT algorithm is mainly developed to address the research problem of spatio-temporal data fusion when

satellite images are limited, specifically when a known pair of images cannot be collected. It is out of the research scope to improve the fusion accuracy with sufficient known image pairs.

In running STARFM, since it requires one more coarse resolution image at the known date for prediction compared to HNN-SPOT, the coarse resolution images were prepared for running the STARFM algorithm. For simulated datasets, the coarse resolution images were prepared by resampling the fine resolution images at the known date to coarse resolution using nearest-neighbor algorithm. For real datasets, the real coarse resolution images (i.e., the MODIS images) were collected at the known dates. The STARFM algorithm was programmed based on [3]. The searching window size was set as 1 coarse pixel so that it is comparable to the window size defined in HNN-SPOT. Since STARFM requires classification data to search for similar pixels, fine resolution classification maps at the known dates were prepared for all study sites. The classification maps were produced using the K-means unsupervised classification algorithm with 4 classes.

To evaluate the fusion results of HNN-SPOT, both quantitative and qualitative measures were adopted. For quantitative assessments, the measures used include the root mean square error (RMSE) to measure the difference between the actual reflectance/radiance and the predicted reflectance/radiance, average difference (AD) indicating the overall bias, correlation coefficient (r) assessing the linear relationship between the actual reflectance/radiance and the predicted reflectance/radiance, and structural similarity (SSIM) [24] comparing the visual structure between the actual image and the predicted image. It is noted that a positive AD indicates overestimation, and SSIM close to 1 indicates a high similarity between the compared images [12]. For qualitative measures, the fusion results were assessed based on visual comparison on the spatial and tonal similarity between the predicted and the true images. Some obvious features on the images were discussed for whether they were captured successfully.

## 4. Results

Referring to the tests done in the study areas using simulated datasets, i.e., study area 1–3, both HNN-SPOT and STARFM produced satisfactory results with low RMSE and AD, and high r and SSIM (see Table 3). The fusion accuracy of HNN-SPOT in these sites was found to be comparable to that of STARFM, even with slightly higher accuracy. Possible reasons for the compatible fusion result by HNN-SPOT compared to STARFM can be due to (1) the similar assumption of the sampling theory between the fine and coarse pixels, and (2) the same usage of moving window in the algorithm, although different mechanisms are applied in the two models. The fusion tasks for study areas 1–3 can be considered as elementary since most of the areas remain unchanged in their spatial details, which indicates that the spatial details of the fine resolution images can be captured over the unchanged areas. Although some spatial features/land uses in study areas 1–3 were changed, the areas of change are large enough to be captured by the coarse resolution images and the fusion algorithms can still use the coarse spectral information to produce satisfactory predictions, although with indistinct object shapes. For example, referring to Figure 7 (study area 1), the vegetation area, where the land cover is consistent, can be favorably predicted with close spatial structure and spectral tone. For the newly built reclamation island near the Hong Kong International Airport, it can still be captured in a relatively indistinct shape although there is no information about its spatial details at the known date. Referring to Figure 8 (study area 2), the unchanged urban structures can be captured satisfactorily and the changing vegetation pattern in the racecourse can still be captured but appears blurred. In Figure 9 (study area 3), the overall spatial details and spectral tone of the cropland were successfully predicted. However, for the changed areas, the fine spatial patterns (e.g., the strips) cannot be produced due to the absence of such details in the fine resolution image at the known date. The fusion accuracies are consistent for HNN-SPOT in study areas 1–3, however, for STARFM, there is a significant decline in accuracy for study area 2. This may imply that HNN-SPOT is stable for different scale factors, while STARFM may not be.

**Table 3.** Quantitative assessments on HNN-SPOT and STARFM (for study area 1-3).

| | RMSE | AD | r | SSIM |
|---|---|---|---|---|
| **Study area 1: Lantau Island and Hong Kong International Airport (scale factor = 16)** | | | | |
| **Reflectance fused by HNN-SPOT** | | | | |
| Blue | 0.0036 | 0.0003 | 0.9645 | 0.9625 |
| Green | 0.0047 | 0.0003 | 0.9493 | 0.9458 |
| Red | 0.0065 | 0.0001 | 0.9363 | 0.9310 |
| **Reflectance fused by STARFM** | | | | |
| Blue | 0.0037 | −0.0002 | 0.9632 | 0.9611 |
| Green | 0.0050 | −0.0003 | 0.9442 | 0.9394 |
| Red | 0.0072 | −0.0006 | 0.9230 | 0.9119 |
| **Study area 2: Hong Kong Island (scale factor = 3)** | | | | |
| **Reflectance fused by HNN-SPOT** | | | | |
| Blue | 0.0122 | −0.0003 | 0.9463 | 0.9446 |
| Green | 0.0098 | −0.0002 | 0.9366 | 0.9340 |
| Red | 0.0085 | −0.0003 | 0.9388 | 0.9363 |
| Reflectance fused by STARFM | | | | |
| Blue | 0.0187 | −0.0009 | 0.8682 | 0.8600 |
| Green | 0.0146 | −0.0001 | 0.8544 | 0.8457 |
| Red | 0.0123 | −0.0002 | 0.8656 | 0.8608 |
| **Study area 3: Werribee South (scale factor = 10)** | | | | |
| **Radiance fused by HNN-SPOT (W/m$^2$srμm)** | | | | |
| Blue | 1.4702 | 0.0122 | 0.9176 | 0.9165 |
| Green | 1.3320 | 0.0080 | 0.9390 | 0.9379 |
| Red | 1.6454 | 0.0066 | 0.9344 | 0.9332 |
| **Radiance fused by STARFM (W/m$^2$srμm)** | | | | |
| Blue | 1.6065 | 0.0336 | 0.8943 | 0.8942 |
| Green | 1.5078 | 0.0476 | 0.9169 | 0.9169 |
| Red | 1.8133 | 0.0675 | 0.9149 | 0.9148 |

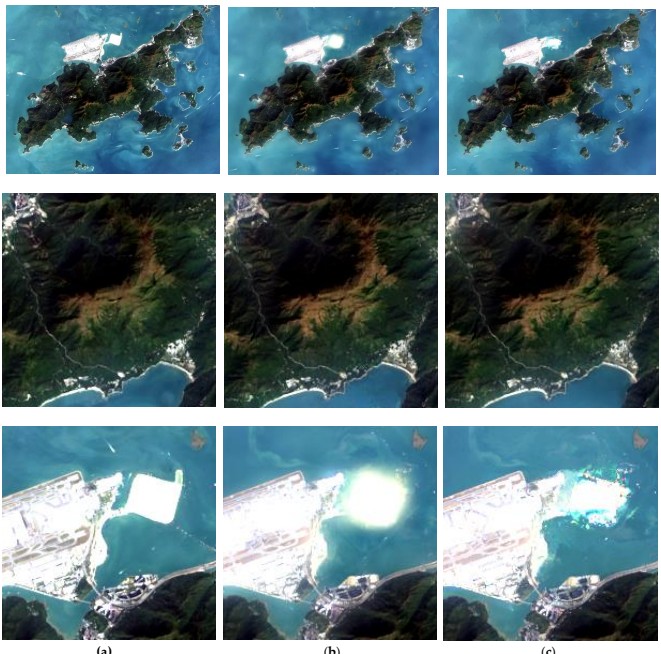

(a)          (b)          (c)

**Figure 7.** Fusion results for Lantau Island and the Hong Kong International Airport on 18 October 2015, zoomed to the unchanged vegetation area (images in the 2nd row) and the newly built artificial island (images in the 3rd row). (**a**) Landsat 8 at 30 m; (**b**) fused image at 30 m by HNN-SPOT; (**c**) fused image at 30 m by STARFM.

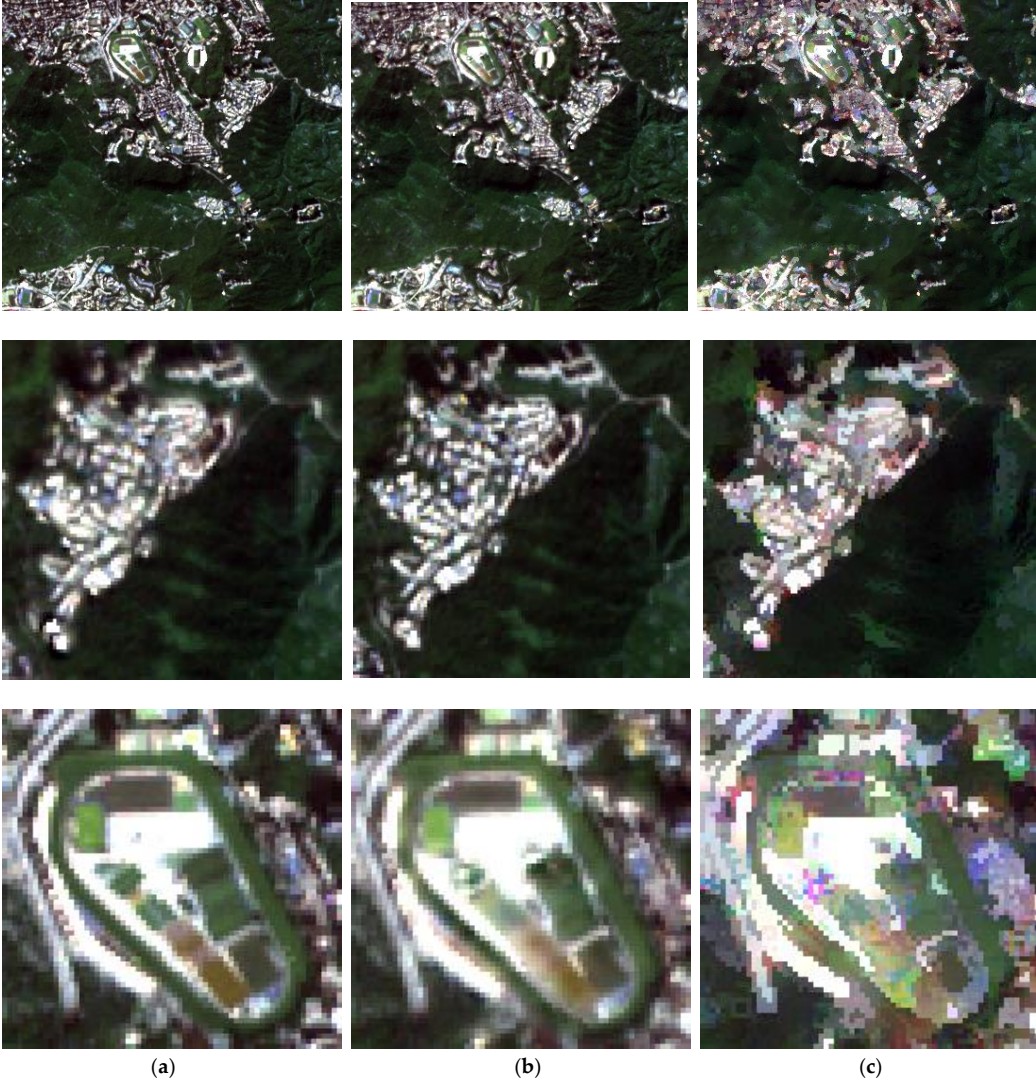

**Figure 8.** Fusion results for the Hong Kong Island on 14 February 2017, zoomed to a piece of the unchanged area with half urban and half vegetation cover (images in the 2nd row) and the racecourse with notable changes (images in the 3rd row). (**a**) Sentinel-2 at 10 m; (**b**) fused image at 10 m by HNN-SPOT; (**c**) fused image at 10 m by STARFM.

For study areas 4 and 5, they are regarded as challenging sites for data fusion, since the spatial structures vary significantly, especially for the Coleambally irrigation area. Here, the changing spatial structures over the Landsat image are relatively small compared with the MODIS image. Therefore, it will be difficult for fusion algorithms to produce accurate results if only the coarse resolution image is available at the prediction date. In addition to the challenge due to abundant change of spatial structures, the inconsistency of sensor characteristics such as different spectral bands and observation angles also pose challenges in the fusion implementations and induce errors in the fusion results. In Table 4, both HNN-SPOT and STARFM show unsatisfactory results with low r and SSIM. This suggests the fusion algorithms cannot predict the spatial structures accurately. Referring to Figure 10, both HNN-SPOT and STARFM cannot produce precise tonal behavior in the irrigation areas. This is due to the assumption of consistent spatial patterns between the known date and the prediction date for both fusion algorithms. Therefore, for HNN-SPOT, it distributes the coarse spectral information of MODIS at the prediction date to the spatial structures of Landsat at the known date, instead of constructing new spatial structures as expected in the Landsat image at the prediction date. The fusion results from both HNN-SPOT and STARFM indicate that the flooded areas were loosely captured without

concrete shape (Figure 11). This problem is also due to the assumption of consistent land cover pattern over the same geographic extent between the known date and the prediction date. Comparing the quantitative measures for HNN-SPOT and STARFM over study areas 4 and 5, STARFM shows more promising results in the Coleambally irrigation area while results of HNN-SPOT are more promising in the lower Gwydir catchment. As suggested by [23], the Coleambally irrigation area is deemed as spatially dynamic and the lower Gwydir catchment is deemed as temporally dynamic. This may suggest that the inclusion of a spatial relationship between the fine resolution image and the coarse resolution image at the known date is more useful in modeling spatial dynamics, but not necessary for the temporal dynamics.

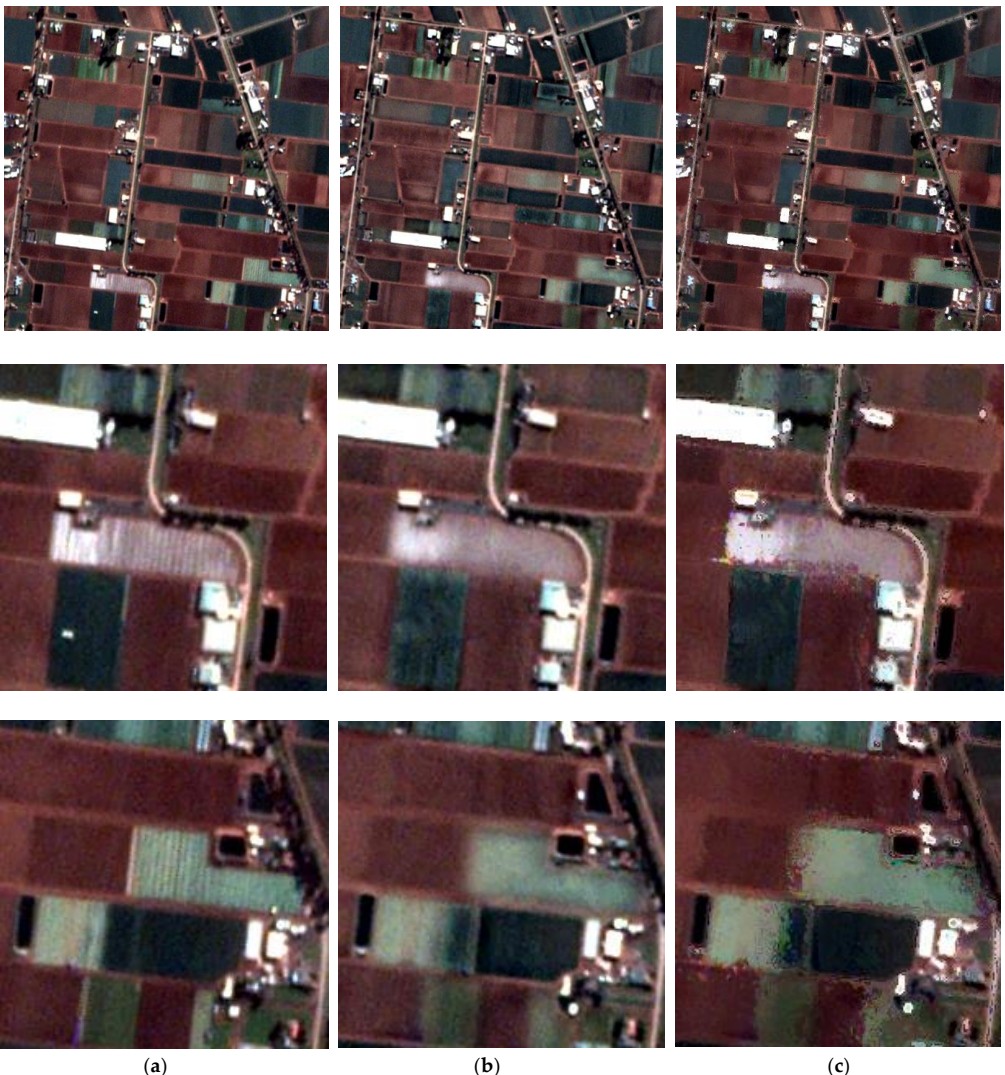

(**a**)　　　　　　　　　　　(**b**)　　　　　　　　　　　(**c**)

**Figure 9.** Fusion results for Werribee South on 31 May, 2018, zoomed to two changing fields (images in the 2nd row and the 3rd row). (**a**) PlanetScope at 3 m; (**b**) fused image at 3 m by HNN-SPOT; (**c**) fused image at 3 m by STARFM.

**Table 4.** Quantitative assessments on HNN-SPOT and STARFM (for study area 4 and 5).

| | RMSE | AD | r | SSIM |
|---|---|---|---|---|
| **Study area 4: Coleambally Irrigation Area (scale factor = 16)** | | | | |
| **Reflectance fused by HNN-SPOT** | | | | |
| Blue | 0.0506 | −0.0494 | 0.4062 | 0.3545 |
| Green | 0.0496 | −0.0469 | 0.3746 | 0.3183 |
| Red | 0.0583 | −0.0518 | 0.4256 | 0.3521 |
| **Reflectance fused by STARFM** | | | | |
| Blue | 0.0440 | −0.0425 | 0.4086 | 0.3740 |
| Green | 0.0405 | −0.0369 | 0.3892 | 0.3581 |
| Red | 0.0476 | −0.0387 | 0.4272 | 0.3918 |
| **Study area 5: Lower Gwydir Catchment (scale factor = 16))** | | | | |
| **Reflectance fused by HNN-SPOT (W/m$^2$srμm)** | | | | |
| Blue | 0.0143 | −0.0039 | 0.6973 | 0.6927 |
| Green | 0.0217 | −0.0065 | 0.6927 | 0.6870 |
| Red | 0.0271 | −0.0077 | 0.6967 | 0.6919 |
| **Reflectance fused by STARFM (W/m$^2$srμm)** | | | | |
| Blue | 0.0169 | 0.0087 | 0.6606 | 0.6527 |
| Green | 0.0235 | 0.0095 | 0.6605 | 0.6497 |
| Red | 0.0294 | 0.0112 | 0.6605 | 0.6525 |

(**a**)　　　　　　　　　(**b**)　　　　　　　　　(**c**)

**Figure 10.** Fusion results for Coleambally irrigation area on 27 April 2002, zoomed to two changing irrigation areas (images in the 2nd row and the 3rd row). (**a**) Landsat at 30 m; (**b**) fused image at 30 m by HNN-SPOT; (**c**) fused image at 30 m by STARFM.

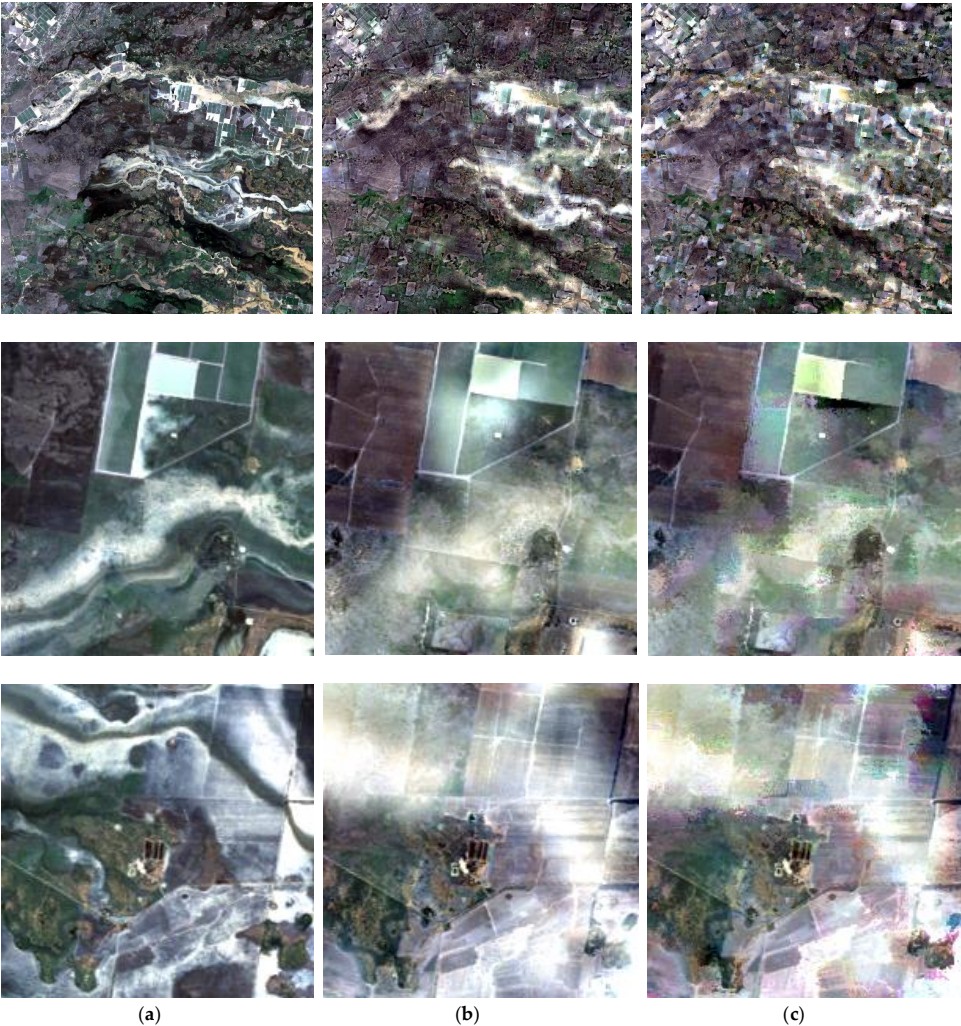

**Figure 11.** Fusion results for lower Gwydir catchment on 12 December 2004, zoomed to two areas covered by water (images in the 2nd row and the 3rd row). (**a**) Landsat at 30 m; (**b**) fused image at 30 m by HNN-SPOT; (**c**) fused image at 30 m by STARFM.

The test results using both the simulated datasets and real datasets show that HNN-SPOT is compatible with STARFM, a benchmark algorithm in spatio-temporal data fusion. The HNN-SPOT algorithm should be useful in efficiently distributing the coarse spectral information at the prediction date to the spatial details, which is assumed unchanged, at the known date. If there are changes in spatial structure which are evident enough to be captured by the coarse resolution image at the prediction date, HNN-SPOT can still produce preliminary estimations for them, although with low details. If there are changes in spatial details which cannot be evidently captured in the coarse resolution image (due to large scale factor), HNN-SPOT fails to produce satisfactory results for the new spatial arrangements. In this regard, HNN-SPOT should be considered as a fusion algorithm for land cover with consistent spatial structures, when observation data are limited to perform more advanced data fusion algorithms.

## 5. Sensitivity Tests

Several parameters were input into the HNN-SPOT algorithm. This section analyzes the influence of different parameter settings on the fusion results. In conducting the sensitivity tests, only the target parameter was varied while others were kept constant. The blue reflectance band of Lantau Island and the Hong Kong International Airport area (study area 1) was used for the sensitivity test.

### 5.1. Varying the Window Size

The window size in HNN-SPOT determines the extent of area to be covered in calculating the correlation coefficient between the fine images at the known date and at the prediction date. Smaller window size indicates that the correlation coefficient is determined using a smaller area extent, and vice versa. To evaluate the impact of different window sizes on fusion results, window sizes from 1 coarse pixel to 3 coarse pixels were tested. In Figure 12, it is shown that the RMSE increases and r decreases as window size increases. This can be explained by the fact that stronger correlation between fine images at the known date and the prediction date is easier to achieve with larger window size. Therefore, when larger window sizes are used, the algorithm loses the rigorous requirement in maintaining fine spatial details. It should be noted that this argument assumes consistent land cover between the known date and the prediction date, and therefore it may not be applicable to areas with significant spatial changes.

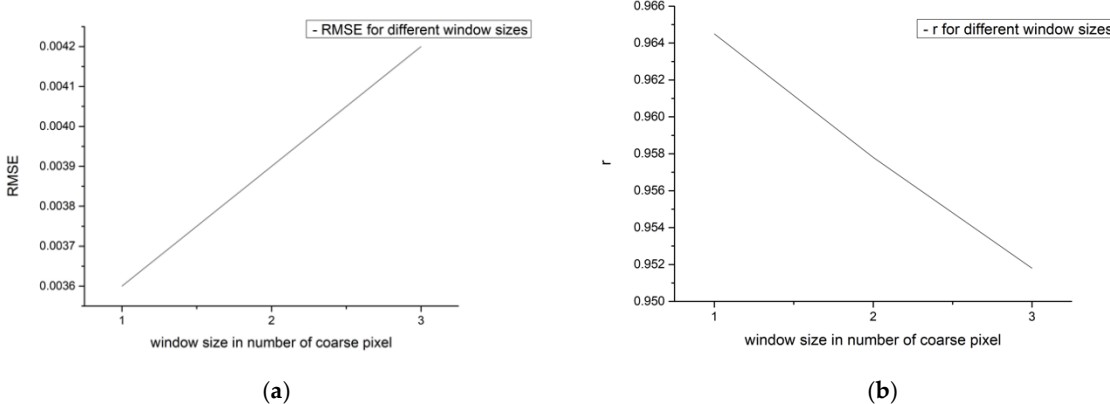

(**a**)                                                                                  (**b**)

**Figure 12.** The (**a**) RMSE and (**b**) r of the fusion results with applying different window sizes (from 1 coarse pixel to 3 coarse pixels) in the HNN-SPOT algorithm.

### 5.2. Varying the Weights $k_1$ and $k_2$

The weights, for the importance of maintaining the fine spatial details ($k_1$) and coarse spectral response ($k_2$), affect the convergence of neurons in the HNN and therefore should be analyzed in sensitivity tests. Different combinations of $k_1$ and $k_2$ have been tested from their low impacts (less weight) to their high impacts (more weight) (Table 5). Figure 13 shows that the combinations of ($k_1 = 0.50$, $k_2 = 1.50$) and ($k_1 = 0.75$, $k_2 = 1.25$) produces the most accurate fusion results for study area 1. When higher weights are given to the component of maintaining the fine spatial details at the known date (i.e., when $k_1$ increases and $k_2$ decreases), the fusion results show more discrepancy. This suggests that maintaining the original spatial arrangement at the known date cannot help in improving the fusion accuracy. This can be due to the varying spatial structures in land cover between the known date and the prediction date, which violates the assumption of consistent spatial pattern (the component assigned with weight $k_1$). Regarding the combinations of $k_1$ and $k_2$ in producing the optimal fusion results, it may imply that the coarse spectral information is comparably more useful in generating reliable fusion results. However, it does not imply that maintaining fine spatial structure is not important because when $k_1$ is too low (i.e., when $k_1 = 0.25$), the discrepancy in the fusion results begins to increase.

**Table 5.** Different combinations of $k_1$ and $k_2$.

| Test Number | $k_1$ | $k_2$ |
|:---:|:---:|:---:|
| 1 | 0.25 | 1.75 |
| 2 | 0.50 | 1.50 |
| 3 | 0.75 | 1.25 |
| 4 | 1.0 | 1.0 |
| 5 | 1.25 | 0.75 |
| 6 | 1.50 | 0.50 |
| 7 | 1.75 | 0.25 |

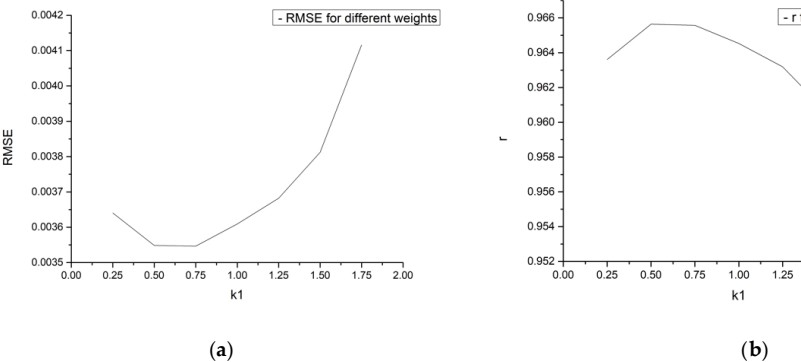

(a)          (b)

**Figure 13.** The (**a**) RMSE and (**b**) r of the fusion results with applying different weights for $k_1$ and $k_2$ in the HNN-SPOT algorithm.

### 5.3. Varying the Threshold of correlation coefficient

In HNN-SPOT, the threshold of correlation coefficient is used to control the requirement of correlation coefficient between the known fine resolution image and the fused image to be predicted, as one of the components to be balanced. Setting $thres_r$ to 1 indicates the assumption of consistent land cover between the known date and the prediction date. When it is defined as below 1, such assumption becomes less strict. In the testing of the requirement on the correlation coefficient between the fine resolution images, the value of $thres_r$ is varied for [1.0, 0.8, 0.6, 0.4, 0.2]. In Figure 14, it shows that the fusion result is the most accurate when $thres_r$ is set to 1.0, where consistent land cover is assumed. Most of the spatial structure in study area 1 remains unchanged and therefore the information of fine spatial details at the known date is useful in prediction. For this reason, fully utilizing the fine spatial details by setting 1.0 to $thres_r$ should produce the most accurate fusion result. Decreasing the value of $thres_r$ uses less fine spatial details information for guiding the prediction, hence resulting in more discrepancies in the fusion results.

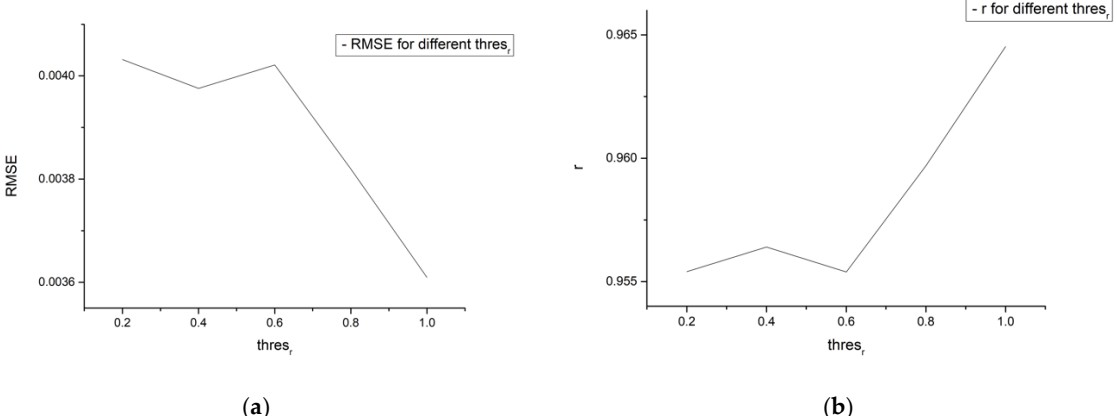

(a)          (b)

**Figure 14.** The (**a**) RMSE and (**b**) r of the fusion results with applying different thresholds for correlation coefficient in the HNN-SPOT algorithm.

## 6. Discussion

HNN-SPOT is developed to tackle the spatio-temporal data fusion problem when observation data are limited. The rationale of HNN-SPOT is based on using one fine resolution image at an arbitrary date and one coarse resolution image at the prediction date. Due to more accurate fusion results generated by HNN-SPOT was observed on the consistent land covers, it is recommended that the two dates used in data fusion should be reasonably close for a higher chance that the land cover remains consistent. The appropriateness of the time period between the known date and the prediction date depends on different criteria such as the extent of the fusion area and the revisit time of the satellite providing fine resolution images. In a normal case, the closer the two dates, the more accurate the fusion results will be, since there will not be large spatial change in the fusion area. However, sometimes this practice will not give better results due to a sudden change in the land cover, for example, flooding and snow fall.

Compared to existing spatio-temporal data fusion algorithms, HNN-SPOT uses only two images and this significantly reduces the data prerequisite. This is advantageous in terms of increasing the applicability of the data fusion technique when observation data are limited. For example, it improves the feasibility of spatio-temporal data fusion in areas with frequent cloud contamination, where it is difficult to collect several clear fine resolution images here. In addition, it enables data fusion between two satellites covering the same area but with different overpass times (where an image pair of fine and coarse resolution images cannot be obtained), and this is the major advantage of HNN-SPOT, compared with other current fusion algorithms. With minimal data input, HNN-SPOT also minimizes some preprocessing errors such as land use classification in STARFM-like algorithms.

Besides of the advantage of data minimization, HNN-SPOT can be easily modified for future improvements owing to the HNN-based architecture. As stated in Equation (1), an energy function is defined by the goals and constraints in a problem, and therefore, the HNN fusion model can be easily modified by adding new goals and appropriately formulated new constraints. In the current formulation of HNN-SPOT, fine spatial detail and coarse spectral response are balanced with equal weight. More research could be conducted in the future to improve the algorithms in HNN-SPOT in order to increase its ability in spatio-temporal image fusion.

Although data minimization stands out as a merit in the development of the HNN-SPOT, its prediction accuracy decreases when more land cover changes occur. For balancing data requirement and prediction accuracy, the accuracy of HNN-SPOT could be improved by including one more fine resolution image in the prediction, although this would increase the data prerequisite. To start with, the HNN-SPOT model is implemented for the fine resolution image before the prediction date and the coarse resolution image at the prediction date, resulting in a fused image (referred to as $Fused_1$). The HNN-SPOT is then implemented again for the fine resolution image after the prediction date and the same coarse resolution image at that prediction date, resulting in another fused image (referred to as $Fused_2$). $Fused_1$ and $Fused_2$ are then combined to predict the final fused fine resolution image through appropriate interpolation. For example, if linear growth is assumed for the area that is fused, a linear time weight can then be applied to $Fused_1$ and $Fused_2$ in estimating the final fused image. The advantage of this method is the prediction of the fused image by referring to the spatial details in two fine resolution images. When there is land cover change at the prediction date and such change still exists in the fine resolution image after the prediction date, this modification of HNN-SPOT can use the fine spatial information on the image after the prediction date. It can therefore predict more accurate fusion results as it does not rely on only the spectral signal in the coarse image at the prediction date (as in the original version of HNN-SPOT). However, when there is land cover change at the prediction date but this change no longer exists in the fine resolution image after the prediction date, such an additionally acquired image will not help in improving the fusion accuracy.

To apply HNN in the field of remote sensing, the developed HNN-SPOT algorithm is a new HNN-based model used for continuous values prediction. Previous applications, for example feature tracking and super-resolution mapping, produce discrete values such as 0 and 1 or class numbers, and are therefore limited to classification-based applications. However, in HNN-SPOT, quantities

with continuous values such as radiance can be predicted. Thus, the development of HNN-SPOT provides a new approach in applying HNN to estimate quantities with continuous values in remote sensing applications.

Limitations in HNN-SPOT include its prediction capability for objects with changing shape or spatial structure, which is lower than that for objects with constant shape or spatial structure. This is due to the formulation that the information of spatial detail relies on the previous fine resolution image. Therefore, there is no reference of spatial structure for areas which undergo substantial change. In order to accurately capture objects which undergo change, fine satellite images of another day can be additionally included in the HNN-SPOT algorithm but more data would then be needed (as described previously). Therefore, future effort will be made to focus on producing more accurate prediction results over areas with land cover change. Another limitation of HNN-SPOT is the processing time, which may be multiples of existing fusion algorithms, due to the iterations involved in the HNN algorithm. However, this problem can be mitigated by using more powerful central processing unit (CPU) or graphics processing unit (GPU) in the computer.

## 7. Conclusions

This paper presents a newly developed spatio-temporal data fusion algorithm, HNN-SPOT, which utilizes the optimization property in the HNN algorithm to assist data fusion. HNN-SPOT uses one fine resolution image at an arbitrary date and one coarse resolution image at the prediction date to synthesize a fine resolution image at the prediction date. This is done by balancing the given information of fine spatial detail and coarse spectral response. Results show that HNN-SPOT is compatible with STARFM, a well-known and currently used data fusion algorithm, although less data are used in HNN-SPOT. HNN-SPOT can generate accurate fusion results for areas with less land cover change, at a low RMSE and high r (>90%). However, the prediction accuracies drop when extensive changes occur spatially (with r around 40%) and temporally (with r around 70%). Therefore, it is recommended to apply HNN-SPOT over the areas with less land cover change, and hence the times between the date when the fine resolution is collected, and the prediction date should be close enough. A possible case to apply HNN-SPOT is to generate time-series high resolution images in a clear day from the geostationary satellites, which usually provide coarse satellite images in high temporal frequency. This is applicable because the land cover usually remains consistent within a day, which meets the assumption of the HNN-SPOT algorithm. Compared with current fusion techniques, HNN-SPOT decreases the data input to two images only. Therefore, it enables data fusion for satellites covering the same area but with different overpass times (e.g. fusion between MODIS and Advanced Spaceborne Thermal Emission and Reflection Radiometer (ASTER) images), which could not be achieved in the past. HNN-SPOT has shown with promising accuracy of fusion applications over areas with consistent land covers. HNN-SPOT brings a new approach to spatio-temporal data fusion with the use of the HNN architecture and it is expected to be improved in the future to increase its applicability in assisting fine-scale environmental modeling and monitoring.

**Author Contributions:** Conceptualization, C.H.F. and M.S.W.; Data curation, C.H.F.; Funding acquisition, M.S.W.; Investigation, C.H.F.; Methodology, C.H.F.; Project administration, M.S.W.; Resources, P.W.C.; Visualization, C.H.F. and M.S.W.; Writing—original draft, C.H.F.; Writing—review & editing, M.S.W. and P.W.C.

**Funding:** This research was funded by part by the grant of Early Career Scheme (project id: 25201614) from the Research Grants Council of Hong Kong; the grant 1-BBWD from the Research Institute for Sustainable Urban Development, the Hong Kong Polytechnic University; and the grant G-YBU3 from the Hong Kong Polytechnic University.

**Acknowledgments:** The authors thankfully express gratitude to the National Aeronautics and Space Administration (NASA) Land Processed Distributed Active Archive Center (LP DAAC) for MODIS; U.S. Geological Survey (USGS) Earth Resources Observation and Science (EROS) Center for Landsat; Planet for PlanetScope; European Space Agency for Sentinel-2 satellite images.

**Conflicts of Interest:** The authors declare no conflict of interest.

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
