# Peer review of "Spatio-Temporal Data Fusion for Satellite Images Using Hopfield Neural Network"

_remotesensing, doi:10.3390/rs11182077_

Round 1

Reviewer 1 Report

In this paper, the authors proposed a new fusion method for combining satellite images based on the Hopfield Neural Network. In general, the paper is well organized and easy to follow. Appropriate reviews of existing methods have been provided and the proposed method has been well evaluated through both comparisons with the existing method and comprehensive sensitivity analysis. However, I do have a major concern about the method. In detail, I am suspicious about the good performance of the proposed fusion method from a statistical point of view. The idea of traditional fusion method is very straightforward: as we have the observed fine-resolution image (say Y) and coarse-resolution image (say X) at time t1, we can build a statistical relationship between Y and X (say Y = f(x)); then, once the coarse-resolution image at time t2 (say X’) is available, we can use the statistical relationship to predict the fine-resolution image at time t2 (say Y’). In comparison, the method here only needs the inputs of Y (fine-resolution image at time t1) and X’ (coarse-resolution image at time t2) in order to predict Y’ (fine-resolution image at time t2). Although the authors claimed that this is the advantage of their method as it requires less input images, it is well known that prediction based on less information usually provides poorer results. Particularly, here the two inputs (Y and X’) are obtained at two different time points, if significant changes have been made to the study area during this period, it is very likely to see poor correlation between Y and X’; how can you use the data to train your model in order to generate reasonable prediction of Y’? Fortunately, the revisit time of satellites is usually short and no significant changes are expected during such a short time of period. But what if we make the time period between t1 and t2 longer than usual, say 30, 60, 180, or 360 days? I believe the traditional fusion method can still generate reasonable results, how about your model?

Other minor comments are listed as follows:

The authors did a good job in comparing their method to the existing one in their result analyses, but I would suggest to put more efforts to discuss why and how the proposed method can produce comparable fusion results, given that the statistical flaw mentioned above. Line 66: “[12] developed” è the author name should be cited here. In general, a raw satellite image may contain many bands (from visible to near infrared, then to far infrared). Does your fusion algorithm deal with each band separately or all in one time? How do you define/identify the moving window? Why do you only consider three sizes of moving window (i.e., 1, 2, and 3)?

Author Response

Please refer to the attached document, thanks

Reviewer 2 Report

The paper by Fung et al. proposed a spatio-temporal data fusion technique using the Hopfield Neural Network. Specifically, the method uses only 2 images to generate a fused image, compared with conventional methods, which employ more than 3 images.

line 13 - 14: “Usually, data which come from two different satellite platforms have different overpass times.”
comment: This is a well-known truth. Readers may fail to see connections between this statement and the research gap. Try to introduce more deduce process before the following sentence.

line 66: “[12] developed the Flexible ……”
comment: Most literature were cited in a proper way, except for some minor issues like this. I recommend the authors check examples on websites (e.g., https://pitt.libguides.com/citationhelp/ieee) and consider to rephrase them.

line 85: “Recently, a spatio-temporal data fusion model ……”
comment: This paragraph seems somewhat abrupt. I didn’t see any obvious discussion on why to introduce CNNs. Hence I recommend the authors elaborate several sentences concerning why employing machine learning methods (e.g., CNNs).

line 301: “For qualitative measures, ……”
comment: The authors should elaborate on the process of how they visually assess the fusion result. What is the criterion?

Figure 2.1
comment: I do not think this is an "architecture".

line 509: “However, it should be noted that ……”
comment: Not entirely clear about how did the authors get this conclusion. This part could be more solid by discussing along with experiment results.

line 564: “Another limitation of HNN_SPOT ……”
comment: The processing time of machine learning methods are much related to hardwares. To make sensce of this sentence, authors could calrify whether they use GPU of CPU, and what is the product model.

While the work of them is acceptable, authors should revise the language to improve readability. I advise them to work with a copyeditor to improve this article.

Author Response

(The authors gave the same response as above.)

Reviewer 3 Report

This is a well written paper, that addresses the important issue of spatio-temporal fusion of satellite imagery. While the authors have provided a thorough explanation of their novel method, there are a few issues that need to be addressed.

Comments:

The HNN-SPOT algorithm is posited in the abstract as an algorithm that can have applications in Environmental Modelling and Monitoring (Lines 29~30). However, in Lines 24~26, the authors note that HNN-SPOT is suitable for areas with little or no land cover change. Monitoring and modelling of the environment is essential because of the changes that take place in land use and land cover. It is therefore contradictory to expect the algorithm to have applicability in an area where its functionality is limited by the very nature of the application, that is, the changes occurring in the environment. Please kindly clarify this. In connection to Comment 1, the test sites selected (Lines 211~220) are described as having undergone some significant spatial change between the reference date and the prediction date. Based on this description, please explain Lines 307~309. This description also contradicts Lines 24~26 regarding the suitability of HNN-SPOT for spatio-temporal fusion. Please kindly clarify this. Please ensure consistency of terms used. Is it ‘spatiotemporal’ or ‘spatio-temporal’? There is no scale, grid, orientation or coordinates provided for all the images (Figures 3.1, 3.2, 4.1, 4.2 and 4.3). This makes it difficult to asses the veracity of the descriptions provided for the test sites. Please kindly georeference the images and add the basic elements of a map or provide coordinates. In lines 438~439, why was blue reflectance chosen? Please provide a reference for Lines 300~302. For Lines 280~282, in essence, all spatio-temporal data fusion algorithms are developed to address this very issue, that is, limited fine resolution images in the temporal dimension. There have been improvements to the STARFM algorithm e.g. the ESTARFM algorithm. In addition, in lines 290~293, major limitation of the STARFM algorithm is highlighted, which further supports the question on why later improved algorithms were not considered for comparison. Why weren't they considered for evaluation? Please elaborate Lines 517~518. The statement implies that the algorithm somehow allows for prediction on dates when images acquired are inundated with cloud cover. Please kindly clarify lines 518~521. This statement may not strictly be accurate since for most of the low-resolution satellites e.g. MODIS, the temporal resolution is daily. This means that at any time when you have a suitable fine resolution image, you will have a coarse resolution image, but the inverse may not be true. Therefore, access to reference and prediction image pairs may not be a great challenge. Please consider revising the grammar in line 524.

Author Response

(The authors gave the same response as above.)

Reviewer 4 Report

Review report on manuscript number ID Remote Sensing-565306 submitted to "remote sensing".

Title: Spatio-temporal data fusion for satellite images using Hopfield Neural Network

Authors: Che Heng Fung and Man Sing Wong

This study proposes a new spatio-temporal data fusion algorithm, the Hopfield Neural Network SPatiotempOral daTa fusion model (HNN-SPOT) is developed, having the ability to use one fine resolution image at an arbitrary date and one coarse resolution image at the prediction date, to predict the fine resolution image at the prediction date using the optimization concept in the Hopfield Neural Network (HNN). Τhe subject is interesting and relevant to the field of this journal.

A number of issues in this paper can be listed as follows:

The language should be improved (Moderate English changes required).

The abstract does not provide the reader with information about the results obtained. It has not any numerical values. It needs to be improved, giving more numerical values for the results.

Avoid lumping references (e.g. [2-12], [2,5,6,8,11]) and similar. Instead summarize the main contribution of each referenced paper in a separate sentence and/or cite the most recent and/or relevant one. The authors should also consider adding there recently published results in the field.

The authors should change all the sentences in the text which are written in the first plural (line 282, 295, 301 and 551). 

The Conclusion is not suitable, should give more useful conclusions. This section presented in brief. This paragraph should be reformed. Should include numerical values for the results.

Author Response

(The authors gave the same response as above.)

Round 2

Reviewer 1 Report

I don't think the authors have properly addressed my major comment (first comment) in the revised manuscript. The authors should at least add more discussions about the potential reasons for the good performance of their developed model, in comparison with traditional methods. Also, there must be some certain cases suitable (or not suitable) for this model, considering the time gap between two images. The authors should clearly describe this in the manuscript.
